# Additive-Manufactured Platinum Thin-Film Strain Gauges for Structural Microstrain Testing at Elevated Temperatures

**DOI:** 10.3390/mi13091472

**Published:** 2022-09-05

**Authors:** Xiaochuan Pan, Fan Lin, Chao Wu, Yingjun Zeng, Guochun Chen, Qinnan Chen, Daoheng Sun, Zhenyin Hai

**Affiliations:** Department of Mechanical and Electrical Engineering, School of Aerospace Engineering, Xiamen University, Xiamen 361005, China

**Keywords:** thin film strain gauge, additive manufacturing, thin film, high temperature

## Abstract

This paper investigates the feasibility and performance of the fabrication of platinum high-temperature thin-film strain sensors on nickel-based alloy substrates by additive manufacturing. The insulating layer was made of a dielectric paste by screen printing process. A 1.8-micron-thick platinum film was deposited directly on the insulating layer. The four-wire resistance measurement method was used to eliminate the contact resistance of the solder joints. Comprehensive morphological and electrical characterization of the platinum thin-film strain gauge was carried out, and good static and dynamic strain responses were obtained, which confirmed that the strain gauge was suitable for in situ strain monitoring of high-temperature complex components.

## 1. Introduction

As one of the most widely used sensors, strain gauges can be used to detect load, pressure, strain, and torque [1,2,3] and have a wide range of applications in structural health monitoring, robotics, electronic skin, and other fields. Strain gauges also have a wide range of applications in flexible electronics, where they can be used to monitor human respiration, movement, and other factors [4,5]. High-temperature thin-film strain gauges (TFSGs) have been widely used to monitor the in situ strain of hot components [6,7,8,9,10,11]. TFSGs are devices capable of measuring strain on the surface of structural components by converting mechanical deformation into electrical signals [12]. They are typically only a few microns thick and can be deposited directly on the components, which work in high temperatures. They do not damage the structure under test, hardly interfere with the flow field characteristics of the surface under test, and due to their negligible mass, they have minimal influence on the inertial vibration of the structure [13,14]. The thin-film sensor has in situ sensing properties, so no bonding process is required, which greatly reduces creep error. In addition, the strain transferability and response speed of thin-film sensors are particularly good, so they are more suitable for dynamic strain testing [9,15,16]. As a high-temperature strain-sensitive material with great research value, metal/alloy materials, such as PdCr and NiCr, show desired linear resistance characteristics versus temperature, which is very important for TFSGs temperature compensation and improving their test accuracy [17]. Metal foil TFSGs are mainly fabricated by sputtering, vapor deposition, and other conditions [17,18]. These coating techniques are difficult to apply to large structures due to the limited size of the processing chamber. In addition, these processes deposit thin films on curved surfaces, resulting in uneven film thickness [6]. Pin-coating and screen-printing, among others, are low-cost and easy-to-implement thin-film processes, but different screens need to be prepared for different patterns. It means their printing flexibility are limited, and less economical for small batch production and multi-step processes [19,20,21,22]. Lithography-based techniques are another traditional method of thin-film sensor fabrication. Although lithographic patterns provide high structural resolution quality, their conformality is insufficient for curved surfaces [12]. Inkjet printing, aerosol jet printing, and direct ink writing (DIW) enable patterning with line widths of hundreds of microns and thicknesses of several microns [1,6,23,24,25,26]. Based on the diversity of printing structures and material choices, DIW provides an alternative fabrication method for the rapid fabrication of thin-film sensors. This method does not require expensive cleanroom facilities and high-end equipment and is proven to be more robust and flexible micro fabrication process [27]. Additionally, this technique enables patterning on three-dimensional surfaces with high structural resolution [28,29,30,31]. Platinum (Pt) is widely known for its high thermal and chemical stability, so it is commonly used in micromechanical devices that operate in harsh environments, such as thermal flow sensors [32] and thermistor temperature sensors [7]. In this study, Pt TFSG was fabricated via DIW based on the Weisenberg’s effect. The DIW technique based on the Weisenberg’s effect is considered one of the most promising patterning methods for solutions with rheological properties [23]. Under the shear thinning effect of the high-speed rotating microneedles, the solution is rapidly transported to the printing needle. Pt TFSG with relatively stable electrical properties was prepared by DIW on GH3536 nickel-based superalloy. The static and dynamic strain responses of Pt TFSG were investigated and tested from 25 °C to 800 °C.

## 2. Materials and Methods

### 2.1. Materials

A high-temperature nickel-based alloy (GH3536, Shenzhen Baishun Metal Materials Co., Ltd., Shenzhen, China), which is commonly used in engineering, was selected as the substrate. A dielectric paste (07HGT90, Shenzhen Sryeo Electronic Paste Co., Ltd., Shenzhen, China) (silicon aluminum calcium oxide) was used as the insulating layer. Pt paste was used for the DIW and silver palladium paste (55H-1805, Shenzhen Sryeo Electronic Paste Co., Ltd., Shenzhen, China) was used to connect the strain grid and Pt leads.

### 2.2. Strain Gauge Fabrication

A schematic diagram of the Pt TFSG fabrication process is shown in Figure 1. First, the GH3536 nickel-based alloy substrate was ultrasonically cleaned sequentially with acetone, deionized water, and alcohol. The dielectric paste was then screen printed on a nickel-based alloy and annealed at 850 °C for 0.5 h. The Pt paste was diluted with butyldiglycol (9:1 mass ratio), followed by patterning using a DIW platform based on the Weissenberg effect. To eliminate the influence of solder joints and leads during the test, a four-wire structure was used. Silver palladium paste was used to connect the Pt strain grid and Pt leads. Finally, the Pt TFSG was sintered at 850 °C for 1 h in an air atmosphere.

### 2.3. Experiment Setup

The strain response of the Pt TFSG was investigated using the cantilever beam method [6], as shown in Figure 2b.The strain of the cantilever is controlled by applying displacement at the free end by a stepper motor. The value of strain ε was calculated using the following equation [33]:(1)$$\varepsilon =\frac{3xyh}{2l^{3}}\;$$
where ε is the strain developed at the location of the TFSGs, y is the deflection of the free end of the substrate, h is the thickness of the substrate, x is the distance from the center of the strain gauge to the point where the load is applied, and l is the length of the substrate.

In this work, one end of the substrate of the strain gauge was clamped and fixed, and the other end was controlled by a stepper motor with an accuracy of 1 micron. By adjusting the stepping distance (y) of the stepper motor, because other parameters (x, h, and l) can be measured with a ruler, the strain on the strain gauge can be calculated.

The indicator of strain gauge sensitivity, also known as the gauge factor (GF), and can be expressed by the following equation:(2)$$\mathrm{GF}=\frac{\Delta R∕R}{\varepsilon }$$
where R is the initial resistance of the TFSG and ΔR is the change in the resistance due to the strain. The high temperature resistance stability of Pt TFSG was tested using a temperature resistance test system, which consists of a tube furnace, a standard thermocouple, data acquisition instruments and a computer (Figure 2a).

The high temperature stability experiment was realized by the temperature resistance test system. The Pt TFSG was placed in the tube furnace, and the tube furnace was heated from room temperature to a specified temperature (400 °C, 500 °C, 600 °C, 700 °C, and 800 °C) at a heating rate of 5 °C/min and then kept for 1 hour, and its high temperature stability was judged by calculating the resistance change rate of the strain gauge.

Compared with the normal temperature strain test system, the high temperature strain test system added a tube furnace. The tube furnace was heated to the specified temperature (400 °C, 500 °C, 600 °C, and 700 °C) at a heating rate of 5 °C/min and maintained for more than half an hour to establish thermal equilibrium. A stepper motor was used to displace the free end of the substrate (applied y) during thermal equilibration, enabling high temperature strain testing.

### 2.4. Characterisation Instruments

The thicknesses of Pt TFSG were measured by a profilometer (Dektak XT). The surface morphology of platinum film was characterized by scanning electron microscopy (SEM, JSM-IT500A). The elements of Pt TFSG were analyzed with energy dispersive spectroscopy (EDS).

## 3. Results and Discussion

### 3.1. Characterizations of the Pt TFSGs

Figure 3a shows the optical image of the printed Pt TFSG. The direct writing process was affected by parameters such as ink viscosity, substrate surface roughness, substrate moving speed, and microneedle rotation speed, so the line width and film thickness of the fabricated samples were different, resulting in different resistances between samples. The resistance of 10 printed platinum sensors was measured at room temperature of 25 °C, and the average resistance was calculated to be 70.5 Ω, the maximum resistance was 75.7 Ω, and the minimum resistance was 65.3 Ω. The Pt TFSG selected for electrical characterization had a resistance of 71.8 Ω at room temperature. The thickness of the strain grid was 1.8 μm and the line width was 620 μm (Figure 3b). Figure 3c shows the micromorphology of the Pt TFSGs. The continuous, uniforms and dense structure ensured that the sensitive film had excellent strain response and electrical conductivity. The EDS analysis (Figure 3d) proved that the main component of the film is Pt.

### 3.2. Strain Testing at Room Temperature

To assess the applicability of Pt as a resistive sensing material, the piezoresistive response characteristics of Pt TFSG were first evaluated by applying different strains at room temperature. Figure 4a shows a plot of the resistance change of Pt TFSG in the strain range of 0 με to 1000 με; 200 με, 400 με, 600 με, 800 με, and 1000 με were sequentially applied to TFSG and kept for 10 s and then unloaded in sequence with a strain amount of 200 με. A recoverable resistance change was clearly observed throughout the loading and unloading process. The distinguishable and stable step signal indicated the excellent static strain response capability of Pt TFSG. The Pt TFSG exhibits a positive GF with an increase in resistance with applied positive strain. Figure 4b depicts the pulsed signals under different strain amounts at a constant strain rate (200 με/s). Resolvable resistance changes indicated the good response of Pt TFSG to strain. Figure 4c shows the dynamic strain response of Pt TFSG at 1000 με. The maximum and minimum values of the pulse signal were almost kept at the same level, indicating the good consistency in dynamic strain response of Pt TFSG. Figure 4d shows the independence of Pt TFSG on strain velocity (strain frequency). Then, 1000 με were applied at strain rates of 20 με/s, 50 με/s, 100 με/s, 200 με/s, and 500 με/s. Despite the different strain velocities, the resistances of Pt TFSG were almost uniform at 1000 με, indicating that Pt TFSG was strain-rate independent.

Figure 5a–d shows the strain response of Pt TFSG under negative strain (compressive strain). Similarly, Pt TFSG exhibited good stability under negative static strain. Under negative dynamic strain, the Pt TFSG exhibited a consistent strain pulse signal. Under negative variable-speed strain signals, Pt TFSG exhibited strain-rate independence. This indicated that Pt TFSG prepared by DIW can be used for dynamic/static strain monitoring and sensing of structural components.

The gauge factor of Pt TFSG could be calculated from the ratio of resistance change rate to strain. In the Figure 6, the slope of the straight line was the GF of Pt TFSG. By calculation, its gauge factor was 1.94. This value was a comprehensive representation of the strain at the Pt TFSG, including the dimensional change and the resistivity change of the strain gauge when it was strained, and the ability of the strain to transfer from the nickel-based alloy substrate to the metal layer was also included [12]. The GF is reported to be in the range of 1.95–2.5 for most metal/alloy strain sensors [12,34,35].

### 3.3. Strain Testing at High Temperature

High temperature resistance stability is also an important capability of TFSG. The resistance as a function of heating and holding time is plotted in Figure 7. For the heating stage, the resistance of Pt TFSG increased with the increase in temperature, showing a positive temperature coefficient of resistance. For the holding stage below 700 °C, there was little change in resistance. For example, at 700 °C for 1 h, the resistance only increased by 0.017%. Since the GF of Pt TFSG was 1.94, the strain error caused by resistance drift at 700 °C for 1 h was only 87 με, indicating that Pt TFSG has good thermal stability. After holding at 800 °C for 1h, the resistance of Pt TFSG increased by 8.2%, and the increase of resistance of Pt film was related to high temperature oxidation [7]. All the above test results demonstrate that the Pt TFSG fabricated by DIW had good thermal stability in the temperature range from room temperature to 700 °C, and the thermal stability at high temperature was the first design principle of high temperature TFSG [6].

The strain responses of Pt TFSGs were performed at 400 °C, 500 °C, 600 °C, and 700 °C using a high-temperature strain testing system. As shown in Figure 8, a strain of 1000 με was applied at a frequency of 0.25 Hz at high temperature and the change in resistance was recorded. It can be observed from the figure that the GF of Pt TFSG also did not decrease significantly up to 600 °C, which was about 1.9. When the temperature was increased to 700 °C, the GF of the TFSG dropped to 1.7. The synchronized changes in relative resistance at the four temperatures indicated the fast response of the Pt TFSG. The relative resistance change is positively linearly related to the strain. Although its high temperature dynamic strain response is not as good as room temperature, it can be seen from the temperature curve in the figure that the resistance fluctuation was mainly caused by temperature fluctuation. This confirms that the DIW-prepared Pt TFSG can be used for dynamic strain monitoring of high temperature up to 700 °C. As shown in Table 1, its high-temperature withstanding capability was better than most TFSGs that have been reported [16,17,36,37,38,39,40,41,42].

## 4. Conclusions

In this paper,

By DIW technology based on Weisenberg’s effect, Pt TFSG with thickness less than 2 μm were fabricated.The strain response of the Pt TFSG at room temperature was well verified with a GF of 1.9 at room temperature.The high temperature stability experiment of the strain gauge showed that its maximum working temperature could reach 700 °C, and the resistance change rate for one hour at this temperature was only 0.017%.The strain test at high temperature showed that the GF of the Pt TFSG was almost unchanged when it was less than 600 °C and decreased to about 1.7 at 700 °C.

Therefore, it can be considered that this strain gauge can be applied to high temperature strain tests below 700 °C. If the printing device for direct writing is integrated with the five-axis processing equipment, it is possible to manufacture thin films on complex curved surfaces, and future work will also focus on this.

## Figures and Tables

**Figure 1 micromachines-13-01472-f001:**
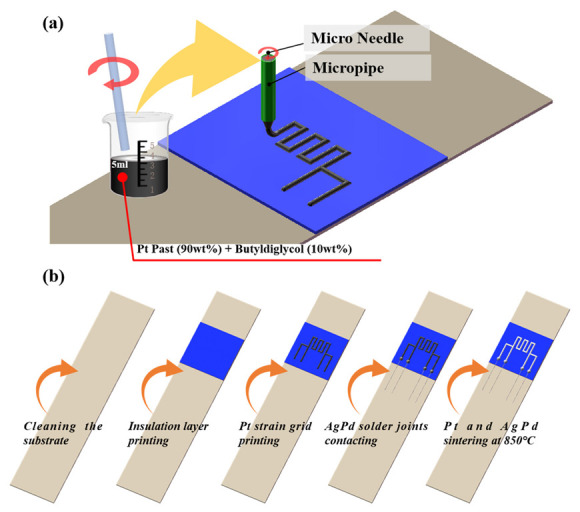
(**a**) Schematic diagram of direct ink writing (DIW) process of Pt paste. (**b**) Process flow diagram of the Pt thin film strain gauge.

**Figure 2 micromachines-13-01472-f002:**
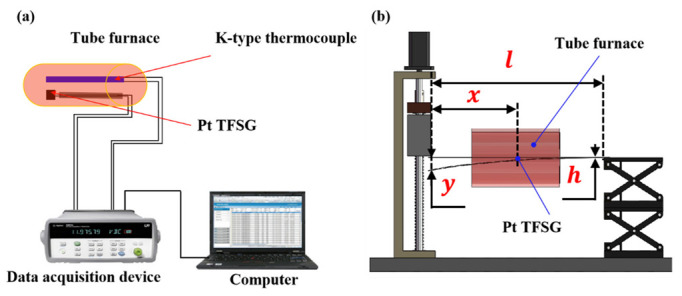
Experiment setups for (**a**) temperature-resistance detection and (**b**) micro-strain detection.

**Figure 3 micromachines-13-01472-f003:**
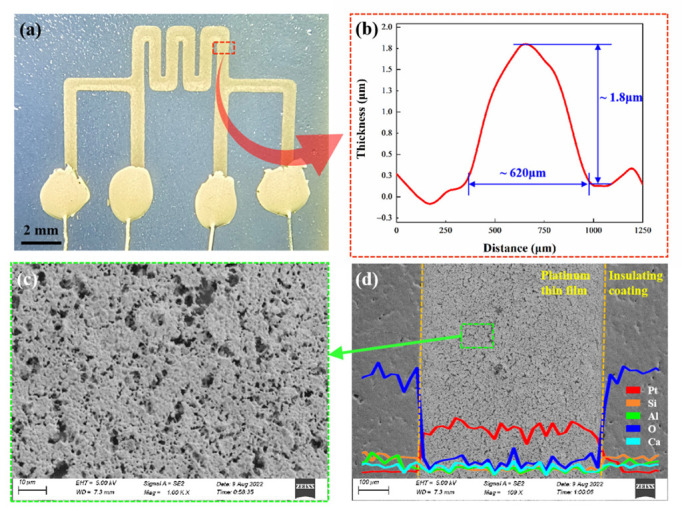
(**a**) Optical image of a printed Pt thin-film strain gauge. (**b**)Thickness of the Pt TFSG. (**c**) SEM images of the Pt sensitive thin film. (**d**) EDS analysis of the Pt TFSG trace.

**Figure 4 micromachines-13-01472-f004:**
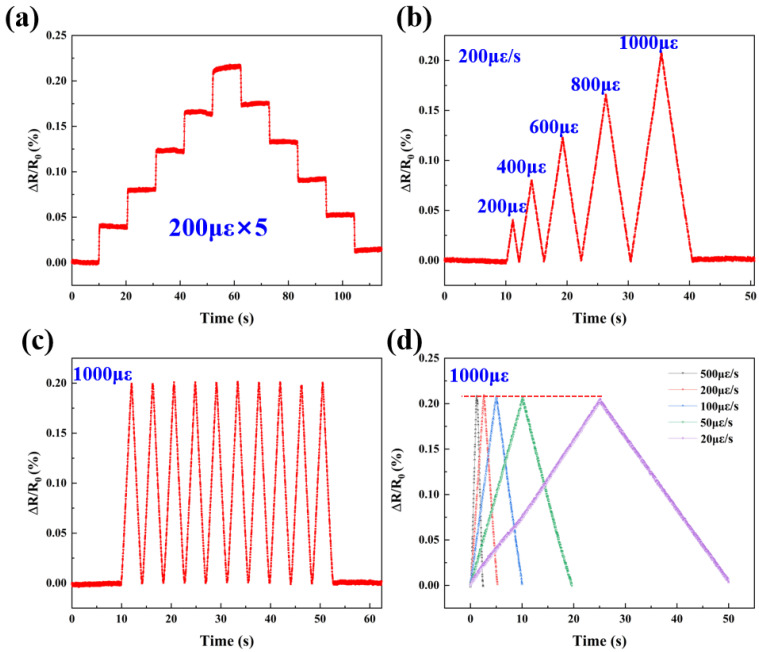
(**a**–**d**) are tensile strain response of Pt TFSGs.

**Figure 5 micromachines-13-01472-f005:**
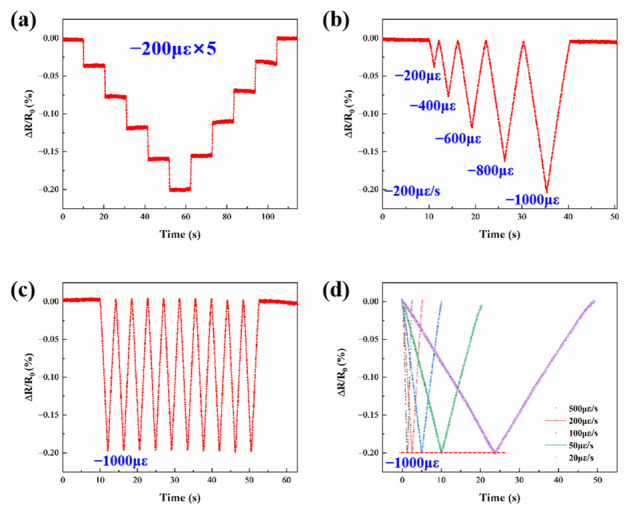
(**a**–**d**) are compressive strain response of Pt TFSGs.

**Figure 6 micromachines-13-01472-f006:**
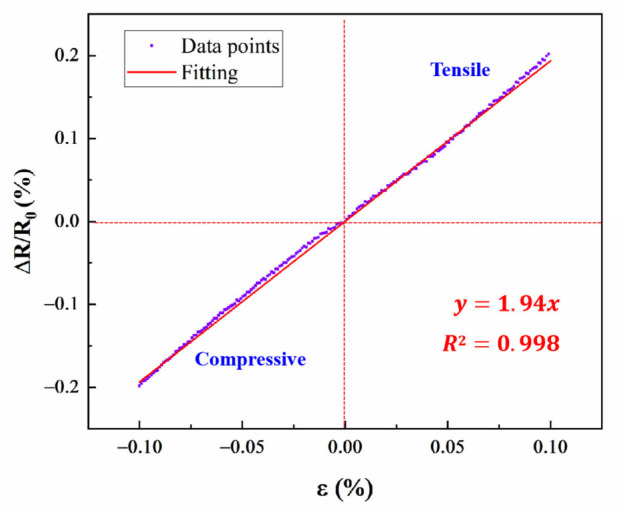
Linear fitting of tensile and compressive strain.

**Figure 7 micromachines-13-01472-f007:**
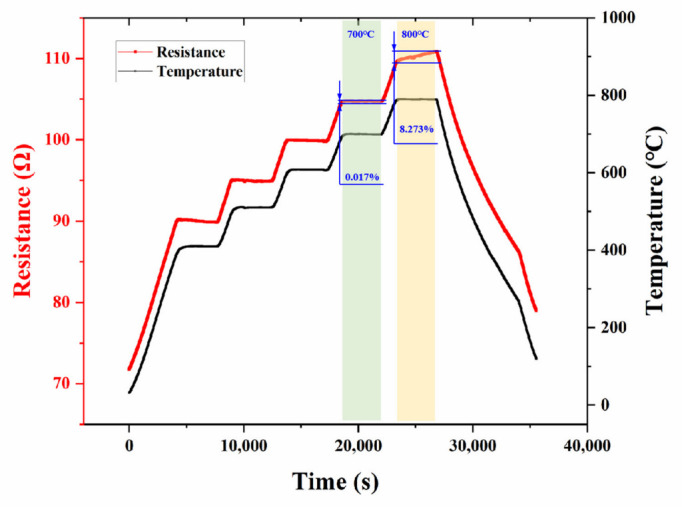
Resistance changes curves of Pt TFSGs at different temperatures.

**Figure 8 micromachines-13-01472-f008:**
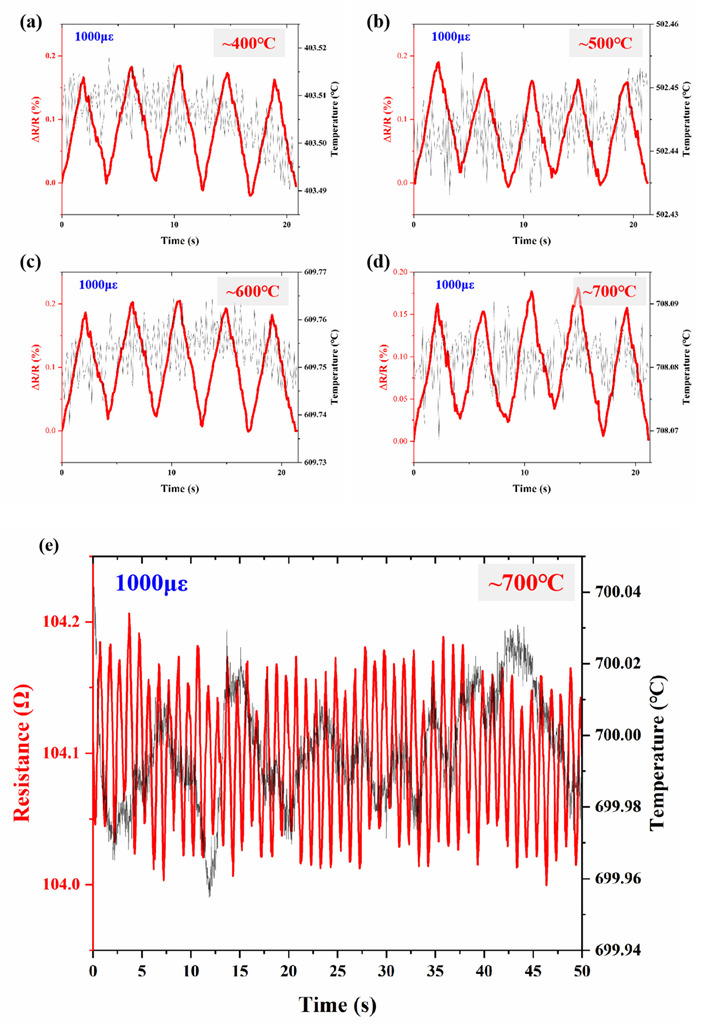
Strain response of Pt TFSGs: (**a**) 400 °C, 5 cycles; (**b**) 500 °C, 5 cycles; (**c**) 600 °C, 5 cycles; (**d**) 700 °C, 5 cycles; (**e**) 700 °C, 50 cycles.

**Table 1 micromachines-13-01472-t001:** Summary of recently released strain gauges.

Material	Maximum Temperature (°C)	Manufacturing Method	GF	Substrate	Reference
Pt	700	DIW	1.7–1.9	Ni-base superalloy	This work
TiAlN_x_O	500	Puttering	2.2–2.5	Sapphire	[37]
AlN/Pt	500	Sputtering	≤4.7	Al_2_O_3_	[16]
Pt	500	Sputtering	1.9–2.5	Sapphire	[38]
Pt/SiO_2_	250	Sputtering	18	Si-wafers	[39]
Pt	440	Sputtering	-	Sapphire	[42]
TiAlN	350	Sputtering	2.5	Sapphire	[36]
Invar36	150	Sputtering	2.5–4.5	Microslides	[40]
NiCr	700 (with a protective layer)	Sputtering	2.5	Ni-base superalloy	[41]

## Data Availability

Not applicable.

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
