# Peer review of "Additive-Manufactured Platinum Thin-Film Strain Gauges for Structural Microstrain Testing at Elevated Temperatures"

_micromachines, 2022, doi:10.3390/mi13091472_

Round 1
Reviewer 1 Report
Overall the paper is technically sound. The paper is not difficult to read but English editing is strongly recommended before publishing. Here are my comments:
1. The paper is about additively manufactured strain sensors but the authors did not provide a thorough review on additively fabricated strain sensors. The authors mentioned pin-coating and screen-printing and their "printing accuracy are limited", but only cited one paper. There are many papers in literature on screen-printed strain gauges and some of them show good printing quality. There are also other AM methods such as inkjet printing, aerosol jet printing, and these methods have been used for strain gauge fabrication.
2. Materials section needs to be improved. The authors did not list where the materials were purchased from.
3. Is the 850C process pyrolysis or sintering? The two words are not interchangeable. What atmosphere was the pyrolysis/sintering performed in?
4. The ordering of Figure 4 and Figure 5 are not the same.
5. How many samples are printed and tested?
6. In figure 7, the authors mentioned that the resistance increase at 800C was due to oxidation. Was the pyrolysis/sintering carried out in inert/reducing atmosphere?
7. The authors did not fully discuss the data tested at high temperatures. It seems there are variations of gauge factor within each test. Why only 5 cycles are shown? Does the gauge factor change at different temperatures? If so what is the gauge factor vs temperature relationship?
Reviewer 2 Report
The current manuscript investigates the performance of additive-manufactured platinum thin-film strain gauges for structural microstrain testing at elevated temperatures. The presented data and results are very interesting and timely, however, there are some issues that should be considered as follows:
- The section that describes the characterization techniques lacks a lot of experimental details.
- Line 111, Figure 2d is not found.
- The scale should be added to Figure 3a.
- The discussion should justify the current study results, and there should be a comparison between the values of obtained results to those published in the literature studies.
- The manuscript should illustrate how the proposed technique could be used to fit complex and irregular surfaces.
- The conclusion should focus on the novelty, main contribution and results using a bullet points style.
Round 2
Reviewer 1 Report
My comments are mostly addressed. The paper needs text editing. Misspellings such as "gage" in the abstract, improper mixed use of present and past tenses, etc. can be easily spotted. The paper overall is not hard to understand but the authors do need to spend some time reading and editing their writings before publishing.
Reviewer 2 Report
The manuscript is improved. However, some issues still need to be considered as follows before being ready for publication:
- Section 2.4 should include more details about the experimental setup and their conditions and parameters, this is a vital point in any academic work.
- Adding a table to compare the current work results to those were obtained in the literature work would be more valuable.
- The conclusion section still needs to be organized using a bullet points style to briefly focus on the main contribution and novelty of the current work.
Round 3
Reviewer 2 Report
The revised manuscript is significantly improved,and all review comments and recommendations are well addressed.